

# Exploring wearable sensors as an alternative to marker-based motion capture in the pitching delivery

Kyle J. Boddy, Joseph A. Marsh, Alex Caravan, Kyle E. Lindley, John O. Scheffey and Michael E. O'Connell

Research and Development, Driveline Baseball, Inc, Kent, WA, USA

## ABSTRACT

**Background:** Improvements in data processing, increased understanding of the biomechanical background behind kinetics and kinematics, and technological advancements in inertial measurement unit (IMU) sensors have enabled high precision in the measurement of joint angles and acceleration on human subjects. This has resulted in new devices that reportedly measure joint angles, arm speed, and stresses to the pitching arms of baseball players. This study seeks to validate one such sensor, the MotusBASEBALL unit, with a marker-based motion capture laboratory.

**Hypothesis:** We hypothesize that the joint angle measurements ("arm slot" and "shoulder rotation") of the MotusBASEBALL device will hold a statistically significant level of reliability and accuracy, but that the "arm speed" and "stress" metrics will not be accurate due to limitations in IMU technology.

**Methods:** A total of 10 healthy subjects threw five to seven fastballs followed by five to seven breaking pitches (slider or curveball) in the motion capture lab. Subjects wore retroreflective markers and the MotusBASEBALL sensor simultaneously.

**Results:** It was found that the arm slot ($R = 0.975$, $P < 0.001$), shoulder rotation ($R = 0.749$, $P < 0.001$), and stress ($R = 0.667$, $P = 0.001$ when compared to elbow torque; $R = 0.653$, $P = 0.002$ when compared to shoulder torque) measurements were all significantly correlated with the results from the motion capture lab. Arm speed showed significant correlations to shoulder internal rotation speed ($R = 0.668$, $P = 0.001$) and shoulder velocity magnitude ($R = 0.659$, $P = 0.002$). For the entire sample, arm slot and shoulder rotation measurements were on a similar scale, or within 5–15% in absolute value, of magnitude to measurements from the motion capture test, averaging eight degrees less (12.9% relative differences) and nine degrees (5.4%) less, respectively. Arm speed had a much larger difference, averaging 3,745 deg/s (80.2%) lower than shoulder internal rotation velocity, and 3,891 deg/s (80.8%) less than the shoulder velocity magnitude. The stress metric was found to be 41 Newton meter (Nm; 38.7%) less when compared to elbow torque, and 42 Nm (39.3%) less when compared to shoulder torque. Despite the differences in magnitude, the correlations were extremely strong, indicating that the MotusBASEBALL sensor had high reliability for casual use.

**Conclusion:** This study attempts to validate the use of the MotusBASEBALL for future studies that look at the arm slot, shoulder rotation, arm speed, and stress measurements from the MotusBASEBALL sensor. Excepting elbow extension velocity, all metrics from the MotusBASEBALL unit showed significant correlations

Corresponding author
Kyle J. Boddy,
kyle@drivelinebaseball.com

to their corresponding metrics from motion capture and while some magnitudes differ substantially and therefore fall short in validity, the link between the metrics is strong enough to indicate reliable casual use. Further research should be done to further investigate the validity and reliability of the arm speed metric.

## INTRODUCTION

Technological advancements in the motion capture field have enabled coaches and athletes to better quantify the locomotor demands of their sport. Marker-based motion capture has been shown in research to be capable of measuring the kinematics and kinetics of a baseball pitch (*Richards, 1999*). The OptiTrack camera system (Natural Motion/ OptiTrack, Corvallis, OR, USA) used in this study has also been shown in research to be comparable to other high-end motion capture systems (*Thewlis et al., 2013*).

Marker-based motion capture, however, requires technical expertise and labor, and can be prohibitively expensive to many coaches and athletes. Inertial measurement unit (IMU) based sensors have been used to quantify human movement and have undergone a lot of technological improvements to become increasingly more accurate.

Inertial measurement unit sensors have been validated in research for joint angle measurements in the lower body (*Leardini et al., 2014*), as well as in the upper body (*Morrow et al., 2017*). IMU sensors have been validated for biomechanical analysis in movement-based areas like gait analysis (*Kavanagh & Menz, 2008*), running kinematics (*Provot et al., 2017*), and swimming biomechanics (*De Magalhaes et al., 2014*). IMU sensors have started to gain popularity in measuring the kinematics of throwers, but validation of such sensors has been limited. Specifically for throwing-based movements, one study placed wearable IMU sensors on the arms and measured kinematic positions to determine whether a cricket bowl qualified as legal or not (*Wixted et al., 2011*). Another study used inertial sensors to determine the peak outward acceleration of several cricket bowlers (*Spratford et al., 2014*).

In baseball, one study used IMU sensors to measure kinematics of youth pitchers, but the study focused primarily on pelvis and torso rotation; the sensor attached to the wrist was only used to identify the timing of the throwing motion's acceleration phase (*Grimpampi et al., 2016*). Another study compared the kinematics of four different pitchers with a five node IMU setup to an optical lab, but relationships were primarily established qualitatively, and only shoulder rotation speed was analyzed with any statistical rigor (*Lapinski et al., 2009*). Additionally, the sportSemble device used in the study is not commercially available, justifying an investigation into more consumer-grade IMU-based sensors.

The MotusBASEBALL unit (Motus, New York, NY, USA) is a popular IMU sensor that purports to measure the biomechanics of a thrower's elbow. The only existing validation of
**Table 1 Participants' descriptive and performance characteristics.**

| 10 Subjects | Height (in) | Weight (lbs) | FB velocity (mph) | OS velocity (mph) |
|---|---|---|---|---|
| Age: 23.8 ± 4.0 | 73.3 ± 0.8 | 206.1 ± 5.5 | 83.8 ± 3.5 | 71.0 ± 3.6 |

Note:
Biological and performance data on the subjects in the study.

the unit comes from *Camp et al. (2017)*, which states that the MotusBASEBALL sensor was evaluated simultaneously with an eight-camera motion capture system. Correlation coefficients ("*r*" values) between measurements with the two systems were found to be "good to excellent" for all measurements, though no supplemental data were provided. A subsequent study used the MotusBASEBALL unit to look at elbow torque and other parameters in pitchers throwing fastballs and off-speed pitches, but did not provide an attempt at possible validation (*Makhni et al., 2018*).

The purpose of this study is to validate the outputs of the MotusBASEBALL sensor, which are arm speed, arm slot, shoulder rotation, and stress, against the OptiTrack motion capture system. The hypothesis was that the joint angle measurements of arm slot and shoulder rotation would be validated as accurate and reliable, while the arm speed and stress metrics might not be as accurate. The hypothesis was more optimistic about the former two measurements because of the past validation research done around IMU sensors in measuring position or joint angles and rotation around one axis, while being more pessimistic about the latter two measurements as arm movement in three separate planes is more difficult to quantify and the inclusion of acceleration in calculating stress and inverse dynamics could likely lead to a propagation of errors through the multiple derivations of the position.

## METHODS

A total of 10 healthy pitchers, all of collegiate or pro-level experience, volunteered to participate in the study: nine threw overhead, one threw sidearm and all were right-handed. Participants were provided a verbal explanation of the study and its risks and were asked to read and sign an informed consent document before testing. The informed consent documents were generated once Hummingbird IRB approved the study and granted ethical approval to carry out the data collection at the author's facilities (Hummingbird IRB #: 2018-10). Testing proceeded once investigators received verbal confirmation and obtained a witnessed legal signature from the athlete. Heights, weights, and ages of the participants were recorded before the beginning of testing (Table 1).

### Testing procedure

Athletes were given as much time as necessary to prepare and warm-up to throw off the pitching mound. Once ready, pitchers were fitted with reflective markers in preparation for the motion capture test. A total of 47 reflective markers were attached bilaterally on the third distal phalanx, lateral and medial malleolus, calcaneus, tibia, lateral and medial femoral epicondyle, femur, anterior and posterior iliac spine, iliac crest, acromial joint, midpoint of the humerus, lateral and medial humeral epicondyle, midpoint of the
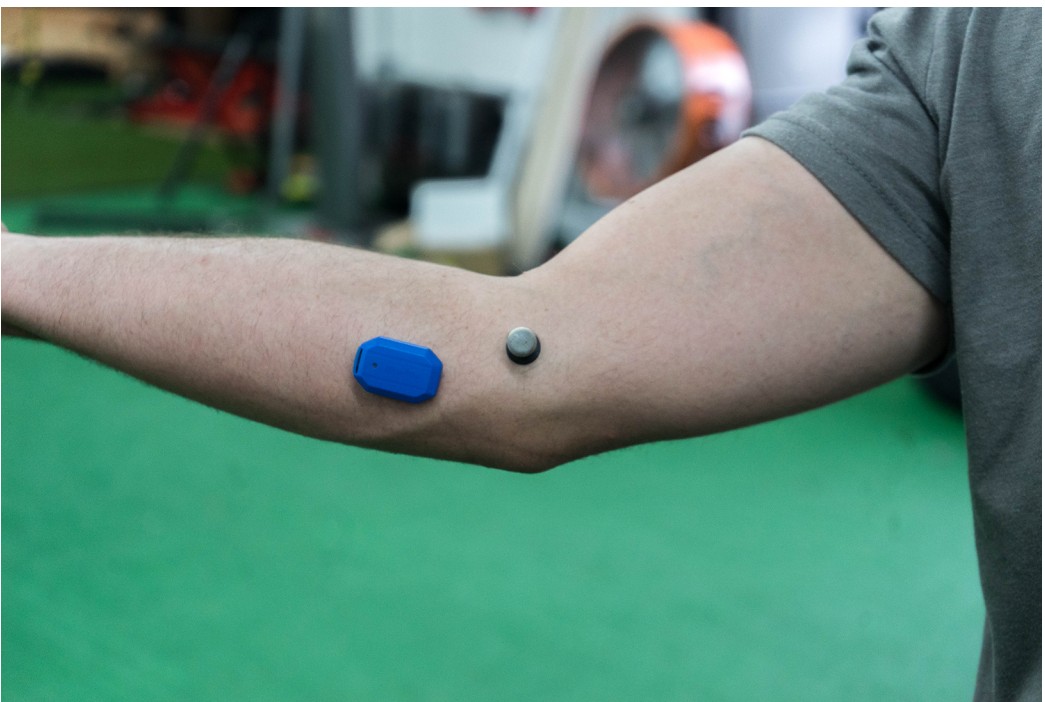

**Figure 1 Placement of the motusBASEBALL sensor on the elbow.** How the motusBASEBALL sensor was affixed to the arm using adhesive instead of the provided sleeve. Photo by Marques Gagner.

ulna, radial styloid, ulnar styloid, distal end of index metacarpal, parietal bone, and frontal bone, as well as on the inferior angle of scapula, C7 and T10 vertebrae, the sternal end of the clavicle, and the xiphoid process.

The motion capture system was calibrated using Motive:Body software (Natural Motion/OptiTrack, Corvallis, OR, USA) and the ground plane was set; the system typically showed one mm or less of mean three-dimensional error, and never exceeded two mm.

The pitchers simultaneously were outfitted with the MotusBASEBALL sensor. The MotusBASEBALL sensor is 38 mm in length, 25 mm in width, 10 mm in height, 6.9 g in weight, and samples at a 1,000 Hz with a three-axis accelerometer (±24 G-sensitivity) and three-axis gyroscope (±4,000 DPS-sensitivity). Further details can be located on Motus Global's Product Specs web page.

Said sensor is typically inserted into a sleeve that the athlete wears, so that the small arrow on the sensor points toward the distal end of the athlete's throwing arm. The sleeve is then worn and adjusted such that the sensor is placed over the flexor bundle of the athlete. The sleeve was not used in this study because of its inability to allow for simultaneous motion capture takes. Instead, the Motus sensor was fixed to the athlete in accordance with the directions on the Motus app, with the designated placer strapping it two finger widths below the medial epicondyle of the inside edge of the athletes throwing forearm using double sided skin-tape to avoid the sleeve causing interference with any of the markers (Fig. 1).

Pitchers then threw fiveto seven fastballs, followed by five to seven off-speed pitches (either curveballs or sliders dependent on each individual's comfort levels), with

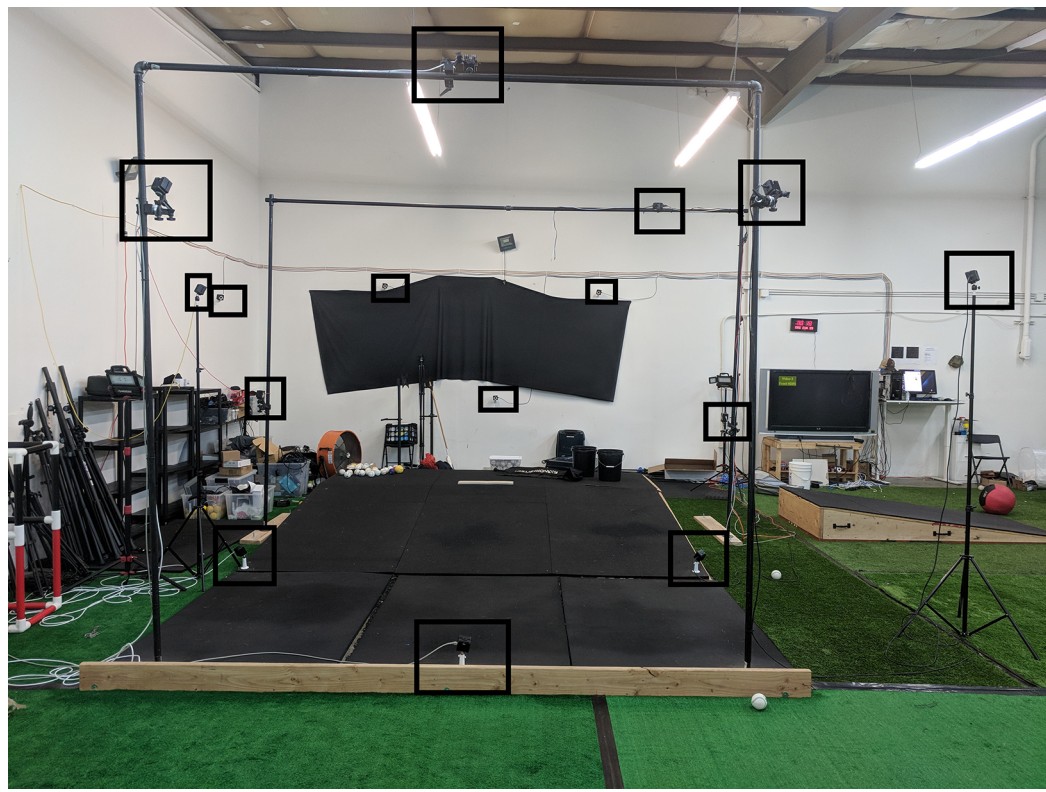

**Figure 2  The motion capture system.** The multi-camera OptiTrack camera system consisting of Prime 13 and Prime 13W cameras, used to evaluate pitcher kinematics and kinetics. Each of the 15 individual cameras identified by squares for clearer black-and-white rendering of the image. Photo by Marques Gagner.                                    

approximately 30–60 s of rest in between throws. All pitches were thrown at a medium effort level. Research has shown that off-speed pitches may result in significant changes to kinetics and kinematics (*Escamilla et al., 2017*; *Fleisig et al., 2006*). For this reason, athletes were asked to throw their preferred off-speed pitch. Fatigue was assumed to be negligible with such a low pitch count.

Throws were made using a five-oz. (142 g) regulation baseball off the mound to a strike zone target (Oates Specialties, LLC, Huntsville, TX, USA) located above home plate, which was 60′ 6″ (18.4 m) away. Testing concluded when the investigators were satisfied they had at least five valid motion capture takes of each pitch type for analysis.

For each trial, ball velocity was measured by a Doppler radar gun (Applied Concepts/Stalker Radar, Richardson, TX, USA). Additionally, for all trials, the three-dimensional motions of the reflective markers were tracked with a multi-camera motion-capture system, sampling at 240 Hz (Natural Motion/Optitrack, Corvallis, OR, USA). This motion-capture system contained a mixture of Prime 13 and Prime 13W cameras, totaling 15 cameras. These cameras were placed symmetrically around the capture volume, approximately 8–12 feet from the center of the pitching mound at varying heights. A total of six cameras were mounted on a truss system in front of the pitcher to avoid collisions (Fig. 2).

Joint centers of the model were estimated based on markers placed on the joint and local coordinate systems (*Dillman, Fleisig & Andrews, 1993*). Position data were filtered using a 20 Hz fourth-order Butterworth low-pass filter, after which kinematics and kinetics were calculated in Visual3D (C-Motion Inc., Germantown, MD, USA). The model was scaled for body size, and inertial properties of the hand, forearm, and upper arm were based on cadaveric data (*De Leva, 1996*). The baseball was modeled as a 0.142 kg point mass at the metacarpal marker until the ball was released, while after release the mass was omitted from the model (*Fleisig et al., 2017*). All kinematic and kinetic values were calculated using the ISB recommended model of joint coordinate systems (*Wu et al., 2002*). In total, 10 kinematic and kinetic values (three position, five velocity, and two kinetic) were calculated and the mean values of each participant's five clearest throws of each pitch type were used (*Escamilla et al., 1998*).

Three position values for the motion capture system were all found at ball release (BR): trunk lateral tilt, shoulder abduction, and shoulder external rotation. Measurements were taken as their local joint angles measured in degrees. The five velocity parameters were taken as the maximum speeds of shoulder internal rotation, shoulder abduction, shoulder horizontal abduction, elbow angular extension, and forearm angular extension, as per the precedents set from the Fleisig model. All velocities were calculated as the rate of change in the joint angle, measured in degrees/second. The two kinetic values calculated were the maximum elbow varus torque and shoulder internal rotation torque, which were measured in Newton meters (Nm).

All MotusBASEBALL data were collected with an iPhone (Apple Inc., Cupertino, CA, USA) and the supplied app, "Motus Throw," which was then manually transferred into labeled spreadsheets for storage and later analysis. The app generated the arm slot, arm speed, arm stress, and shoulder rotation metrics. Arm slot was reported as taken at BR while arm speed was taken at the peak value slightly after BR; the arm stress and shoulder rotation measures were dependent on the athlete's max external rotation.

## Statistical analysis

The data metrics were analyzed as both a total sample of 20 pitches and two separate equal-sized groups classified by the type of pitch: fastballs (10) and off-speed pitches (10). Each pitch was an average of the five pitches analyzed by each of the two systems in question. Anticipating a difference in the scale of the respective magnitudes for the two systems, the statistical analyses centered on a correlation test based around Pearson's product moment of correlation coefficient and an $n-2$ number of degrees of freedom. The correlation test was used to test the hypothesis of a linear relationship between the set of metrics obtained for each of the two systems. Statistical significance was based on a default alpha value of 0.05.

In order to create measurement analogues between the motion capture trial and the MotusBASEBALL metrics, additional calculations were done. Corrections to the metrics were done following Motus's guidelines which were communicated via email by representatives from Motus; those corrections follow below.

Arm slot (Motion Capture system) was taken as the sum of the lateral trunk tilt and shoulder abduction at BR. shoulder rotation was measured as the maximum amount of shoulder external rotation measured in the global coordinate system. MotusBASEBALL's arm speed metric, which was taken from the MotusTHROW app, was compared to elbow extension velocity and shoulder internal rotation velocity, which are the most common standards for measuring arm speed. Per Motus's recommendation, arm speed was also compared to the magnitude of the resultant angular velocity of the shoulder, which is compromised of the following components: the square root of the sum of the squares of shoulder abduction velocity, $\omega_{Sa}$, shoulder horizontal abduction velocity, $\omega_{Sha}$, and shoulder internal rotation velocity, $\omega_{Sir}$. $\sqrt{\omega_{Sa} + \omega_{Sha} + \omega_{Sir}}$.

In addition, the angular velocity of the forearm extension as taken on the motion capture system as another arm speed metric to use based on Motus defining their arm speed metric as the "resultant angular velocity of the forearm segment." MotusBASEBALL stress was compared to elbow varus torque and shoulder internal rotation torque, which are the two most commonly addressed kinetic markers in pitching research. All torque metrics were in Nm.

First, the descriptive metrics (means and standard errors of means) for the whole group and subgroups for all the marker-based biomechanics measurements and MotusBASEBALL measurements were outlined and recorded. Differences in magnitudes across the two systems were rendered as both absolute and relative errors, where the relative error was measured as the absolute error divided by the total value of the motion capture criterion system. Then these metrics were matched together across paired results (each subject having been recorded on the two separate systems), and had both their Pearson correlation coefficient $\rho$ calculated along with its 95% confidence interval and its associated $P$-value, following a Student's $t$-test distribution. The correlation test posits the hypothesis of there being a significant linear association vs the null hypothesis of there being no correlation, or $\rho = 0$. Differences in magnitudes between the systems are noted in the text as both absolute differences and relative percentage differences between the two systems, divided by the motion capture criterion system. In addition, Bland–Altman plots were used for each fastball and off-speed metric comparison to investigate the reliability of the two metrics despite their frequent differences in absolute magnitudes. All the aforementioned statistical analysis was performed using the program open-source statistical program R (*R Core Team, 2018*).

## RESULTS

The results for the three separate groups are displayed in Tables 2 and 3. As is somewhat intuitive given the nature of the more similar sub-populations, the correlation coefficient is higher within said smaller groups, due to the smaller sample sizes and subsequent degrees of freedom. The fastball group found significant associations between four of the metrics (arm slot, shoulder rotation, and the second and third arm speed metrics), while the off-speed group found significant associations between six metrics (arm slot, shoulder rotation, the second and third arm speed metrics, and both stress metrics). Confidence intervals were included to give a clearer picture of the correlation's reliability

**Table 2 Averages of the metrics taken from motion capture analysis compared with the corresponding metrics from MotusBASEBALL.**

| Group | All | | Fastball | | Off-speed | |
|---|---|---|---|---|---|---|
| **Sample size** | **20** | | **10** | | **10** | |
| Metric | Motion capture | MotusBASEBALL | Motion capture | MotusBASEBALL | Motion capture | MotusBASEBALL |
| Arm slot (deg) | 62 ± 3 | 54 ± 8 | 63 ± 5 | 53 ± 8 | 61 ± 5 | 54 ± 5 |
| Shoulder rotation (deg) | 167 ± 2 | 158 ± 5 | 167 ± 3 | 156 ± 5 | 168 ± 3 | 157 ± 3 |
| Arm speed–elbow extension speed (deg/s) | 2,404 ± 38 | 925 ± 24 | 2,398 ± 49 | 945 ± 33 | 2,410 ± 61 | 935 ± 20 |
| Arm speed–shoulder internal rotation speed (deg/s) | 4,670 ± 130 | 925 ± 24 | 4,648 ± 178 | 945 ± 33 | 4,692 ± 199 | 935 ± 20 |
| Arm speed–shoulder velocity magnitude (deg/s) | 4,816 ± 120 | 925 ± 24 | 4,795 ± 167 | 945 ± 33 | 4,838 ± 181 | 935 ± 20 |
| Stress–varus torque (Nm) | 106 ± 4 | 65 ± 3 | 103 ± 5 | 62 ± 2 | 110 ± 6 | 64 ± 2 |
| Stress–shoulder IR torque (Nm) | 107 ± 4 | 65 ± 3 | 104 ± 5 | 62 ± 2 | 111 ± 6 | 64 ± 2 |

Note:
A comparison of the motion capture system using high-precision OptiTrack cameras compared with the metrics the motusBASEBALL unit provides.

**Table 3 P-values and correlations with confidence intervals for metric comparisons.**

| Group | All | | | Fastball | | | Off-speed | | |
|---|---|---|---|---|---|---|---|---|---|
| **Sample size** | **20** | | | **10** | | | **10** | | |
| Metric | *P*-Value | *R* | *R*: C.I. | *P*-Value | *R* | *R*: C.I. | *P*-Value | *R* | *R*: C.I. |
| Arm slot | <0.001* | 0.975 | [0.94–0.99] | <0.001* | 0.978 | [0.91–0.99] | <0.001* | 0.974 | [0.89–0.99] |
| Shoulder rotation | <0.001* | 0.749 | [0.46–0.89] | 0.022* | 0.71 | [0.15–0.93] | 0.007* | 0.784 | [0.30–0.95] |
| Arm speed–elbow extension speed | 0.207 | 0.295 | [−0.17–0.65] | 0.341 | 0.337 | [−0.37–0.80] | 0.413 | 0.292 | [−0.41–0.78] |
| Arm speed–shoulder internal rotation speed | 0.001* | 0.668 | [0.32–0.86] | 0.010* | 0.762 | [0.25–0.94] | 0.045* | 0.643 | [0.02–0.91] |
| Arm speed–shoulder velocity magnitude | 0.002* | 0.659 | [0.31–0.85] | 0.017* | 0.727 | [0.18–0.93] | 0.041* | 0.651 | [0.04–0.91] |
| Arm speed–forearm velocity magnitude | 0.309 | 0.239 | [−0.15–0.66] | 0.446 | 0.322 | [−0.43–0.77] | 0.273 | 0.365 | [−0.39–0.79] |
| Stress–varus torque | 0.001* | 0.667 | [0.32–0.86] | 0.077 | 0.583 | [−0.07–0.89] | 0.011* | 0.759 | [0.66–0.83] |
| Stress–shoulder IR torque | 0.002* | 0.653 | [0.30–0.85] | 0.094 | 0.557 | [−0.11–0.88] | 0.010* | 0.763 | [0.26–0.94] |

Notes:
Statistical analysis of the comparisons between the motion capture system and the motusBASEBALL unit.
* Indicates that the metric was found to be statistically significant at a $P < 0.05$ value.

and confirm that the significant correlations indicate some degree of positive linear relationship. Bland–Altman plots were generated below in Figs. 3–6 for analysis of the different measurement systems and their subsequent reliability. Their reliability appears to be quite high as the individual data points all fell within the confidence intervals of the differences between the systems' magnitudes for the majority of the metrics, and no one metric had more than a single point outside of said confidence intervals.

## DISCUSSION

Arm slot was found to be near perfectly correlated across all groups, though MotusBASEBALL's arm slot was roughly 7–10 degrees lower than the results from our motion capture system, with a relative error of 12.9%.

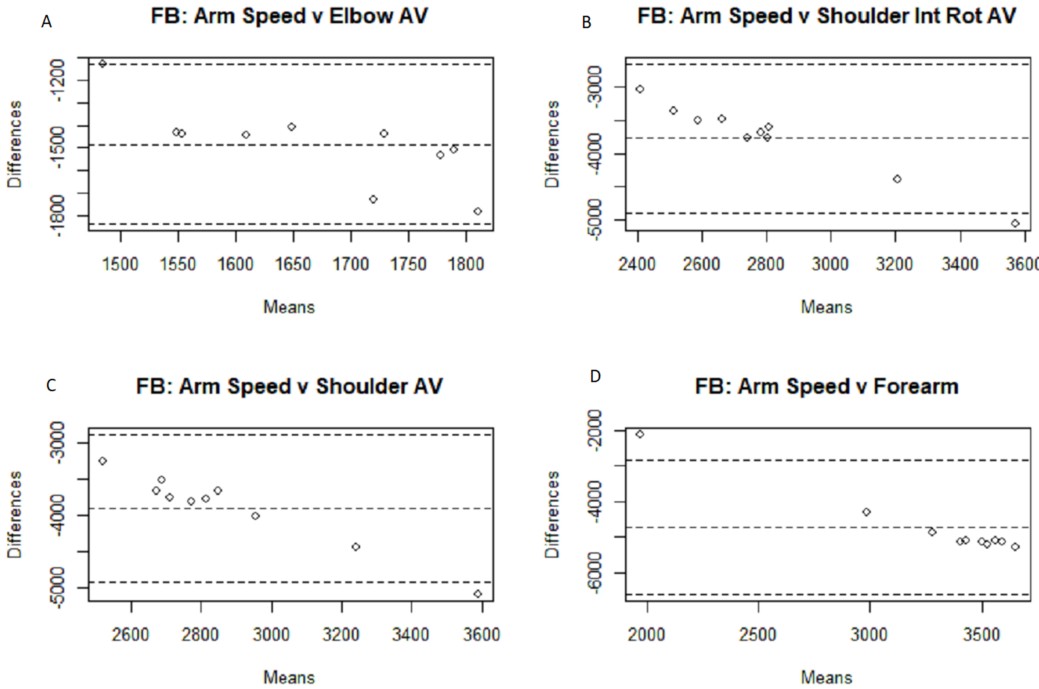

**Figure 3 Bland–Altman plots for fastball arm speed comparisons.** Bland–Altman plots among the fastball pitches for all four motion capture measurements compared to the Motus arm speed metric: elbow angular velocity (A), shoulder internal rotational velocity (B), shoulder angular velocity (C), and forearm extension velocity (D).

Shoulder rotation was also strongly correlated between the two systems. On average the shoulder rotation measured by MotusBASEBALL was nine degrees, or 5.4%, lower than what the motion capture system detected for the total group.

Arm speed from MotusBASEBALL showed strong correlations to both shoulder rotation speed metrics, but no correlation to elbow extension speed or the forearm extension. This could be due to the fact that the MotusBASEBALL sensor is placed very close to the elbow joint, so movement of the forearm caused by elbow extension is much less detectable due to the shorter lever arm that it detects rotation from.

The numerical difference between the two systems is fairly substantial. Average MotusBASEBALL arm speed, which was 925 deg/s, was dramatically lower than the measured shoulder internal rotation speeds and magnitude of both shoulder rotational velocities and forearm velocities, which were 4,670, 4,816, and 5,744 deg/s, respectively, to go along with relative differences of 80.0%, 80.8%, and 83.9%. It is also worth noting that the arm speed metric that MotusBASEBALL outputs in the app is different than the metric that is in their web-based portal. Because MotusBASEBALL's arm speed metric in the app would scale linearly to the metric in the portal, it follows that the comparison of motion capture arm speed metrics to the arm speed in the app would still be reliable.

Both comparisons to MotusBASEBALL's stress metric were significant. Both stress measurements (from MotusBASEBALL and from motion capture) were shown to be consistent across the holistic sample of subjects. Kinetics calculations are heavily

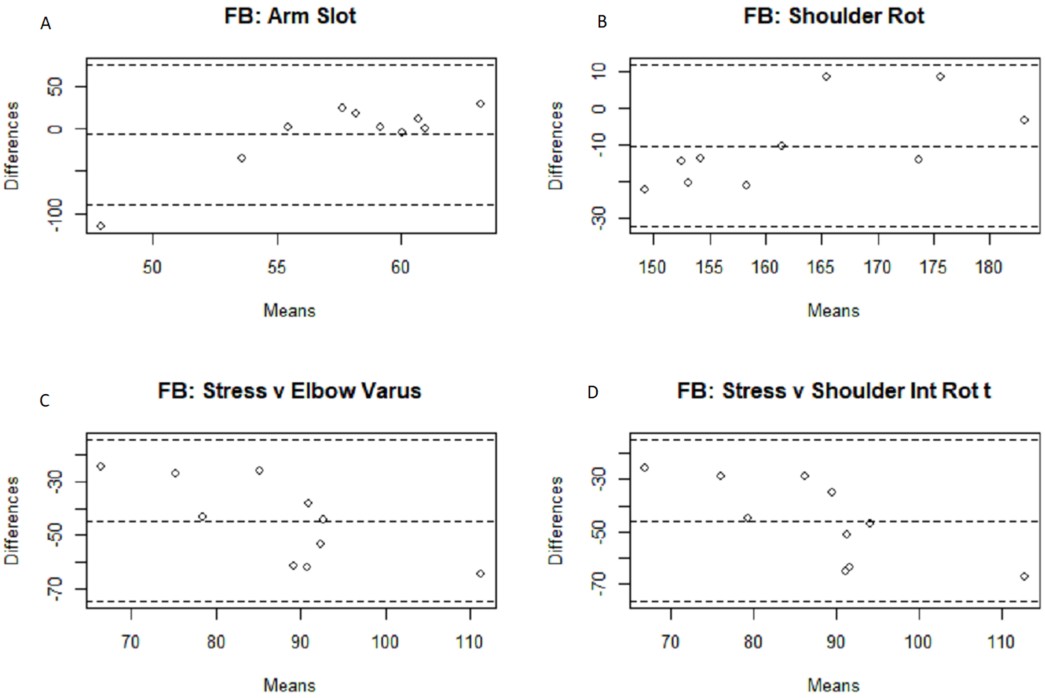

**Figure 4 Bland–Altman fastball arm slot, shoulder rotation, and arm stress comparisons.** Bland–Altman plots among the fastball pitches for the arm slot (A), shoulder rotation (B), and arm stress (against elbow varus torque (C) and shoulder internal rotation torque (D) Motus comparisons.

dependent on the athlete's height and weight, along with the weight of the ball (*Feltner & Dapena, 1986*). Motus has stated that their calculation also takes these factors into consideration and are part of the inputs required to use the MotusBASEBALL sensor. The fact that those inputs are considered could explain part of the statistically significant correlation between the two-stress metrics. Conversely, while the stress correlation exists for the whole sample and the off-speed sub-sample, it is not significant for the fastball sample; potential variables that could explain the disparity in correlations include the differences across the systems in marker placement and inertial parameters set in their respective algorithms.

Because the numerical outputs from the MotusBASEBALL unit are noticeably different from the outputs from marker-based motion capture outputs, which is the gold standard of biomechanical analysis, MotusBASEBALL's best use may be in relative comparisons of the same athlete. This gap in absolute value potentially stems from the difference in measurement units the two systems use; as the Bland–Altman plots above show, the majority of the data points fall within the 95% confidence intervals for all eight metric comparisons in both the fastball and off-speed populations: the only exceptions being a solo arm slot data point for both off-speed and fastball pitches, and a solo data point for the fastball metric comparison of Motus's arm speed and MoCap's shoulder internal rotation angular velocity. Nevertheless, these findings, while supporting the reliability of the Motus metrics, fail to validate them as

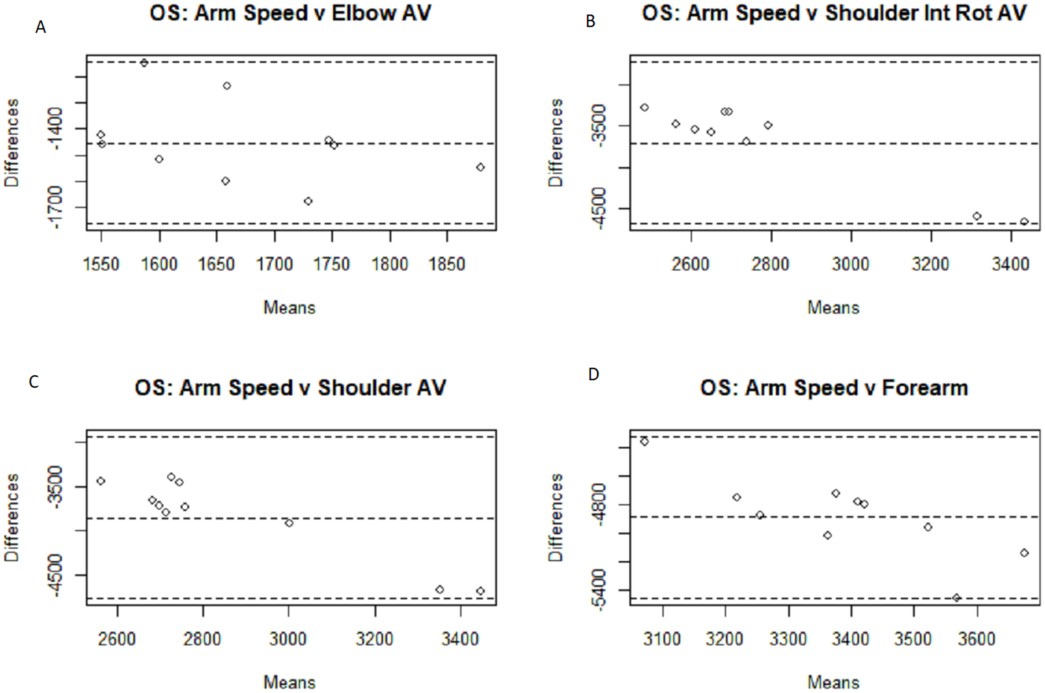

**Figure 5 Bland–Altman plots for off-speed arm speed comparisons.** Bland–Altman plots among the off-speed pitches for all four motion capture measurements compared to the Motus arm speed metric: elbow angular velocity (A), shoulder internal rotational velocity (B), shoulder angular velocity (C), and forearm extension velocity (D).

validation in research, by definition, necessitates the magnitude of the scale to be confirmed as accurate.

These differences in magnitudes are likely in large part from the aforementioned escalating error that stems from IMU sensors attempting to measure movement in three planes and correctly quantify acceleration, the second derivative of position with respect to time. Nevertheless, there are multiple instances of concurrent technologies having significant correlations, and by extension acceptable reliability, while exhibiting numerical differences in absolute magnitude that impede its validity (*O'Donnell et al., 2018*). In addition, there is also a specific history in the world of baseball player development in using technology that may be highly reliable while measuring outcomes on different scales of magnitude, like the tachistoscope test correlating with a player's batting average (*Reichow, Garchow & Baird, 2011*).

In addition, MotusBASEBALL has shown to be internally consistent when used by the same athlete as evidenced by the subjects' individual coefficient of variation scores on their five Motus-recorded throws, which makes it an efficient tool for noting significant changes to an athlete's mechanics (Table 4).

While the MotusBASEBALL unit cannot replace the gold-standard of motion capture, it has a significant advantage in that it can be used in live competition and practice situations without serious preparation. The MotusBASEBALL unit is likely best applied by laypeople, coaches, and those who do not have regular access to a sophisticated motion capture system, or the time to implement said analysis.

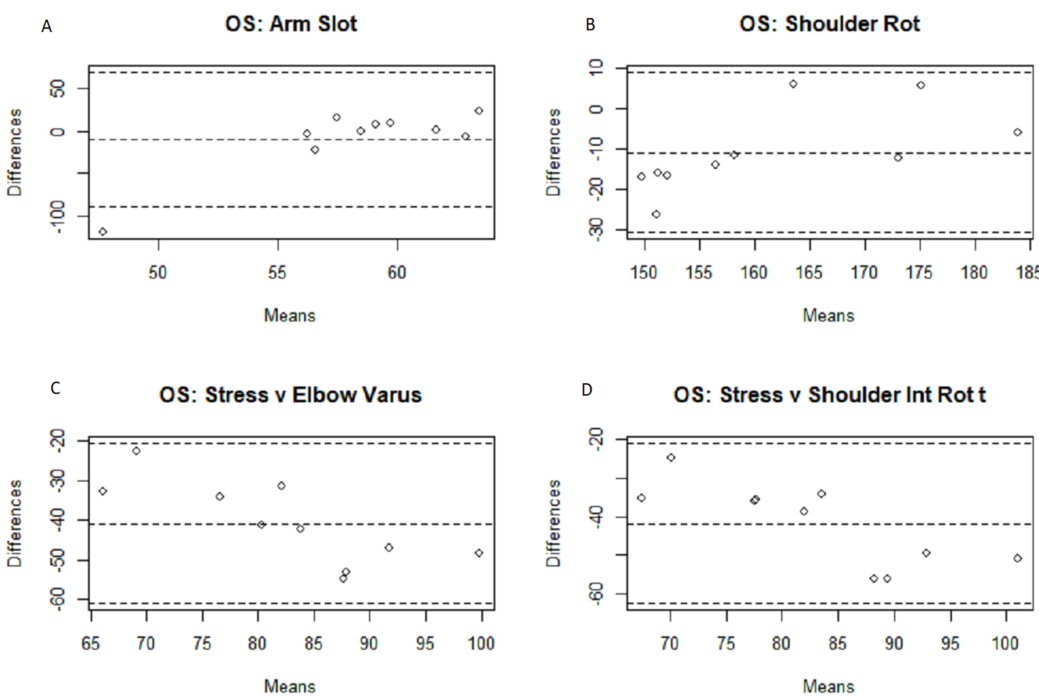

**Figure 6 Bland–Altman off-speed arm slot, shoulder rotation, and arm stress comparisons.** Bland–Altman plots among the off-speed pitches for the arm slot (A), shoulder rotation (B), and arm stress (against elbow varus torque (C) and shoulder internal rotation torque (D) Motus comparisons.

**Table 4 Coefficient of variation for MotusBASEBALL metrics by individual athletes.**

| Athlete | Fastball pitches | | | | Off-speed pitches | | | |
|---|---|---|---|---|---|---|---|---|
| | Arm slot (%) | Shoulder rotation (%) | Arm speed (%) | Stress (%) | Arm slot (%) | Shoulder rotation (%) | Arm speed (%) | Stress (%) |
| 1 | 4.28 | 1.22 | 5.77 | 2.18 | 2.99 | 1.36 | 1.87 | 1.53 |
| 2 | 4.89 | 1.61 | 10.96 | 5.10 | 3.90 | 1.68 | 10.40 | 3.37 |
| 3 | 8.44 | 2.62 | 8.52 | 3.86 | 5.99 | 1.62 | 16.44 | 10.37 |
| 4 | 6.35 | 2.47 | 10.19 | 6.58 | 5.50 | 0.95 | 4.24 | 10.57 |
| 5 | −9.32 | 0.84 | 5.19 | 10.16 | −16.67 | 0.82 | 3.00 | 10.68 |
| 6 | 5.68 | 1.39 | 9.90 | 10.01 | 17.89 | 2.11 | 8.74 | 5.83 |
| 7 | 3.33 | 1.82 | 4.84 | 12.00 | 4.08 | 1.39 | 2.32 | 3.93 |
| 8 | 7.84 | 1.03 | 10.12 | 2.37 | 9.04 | 1.59 | 10.72 | 13.93 |
| 9 | 3.31 | 1.68 | 6.13 | 6.25 | 2.52 | 0.97 | 8.31 | 6.75 |
| 10 | 3.33 | 1.79 | 9.04 | 7.38 | 2.88 | 1.69 | 2.77 | 2.01 |

**Note:**
An athlete-by-athlete analysis of the coefficient of variation scores for all five throws across all Motus-generated metrics.

## Limitations

There are a few noteworthy limitations to this study. As mentioned previously, the more commercial sleeve was not used to place the sensor. Using a sleeve would have prevented the ability to take motion capture takes as the markers could not have been placed

on the sleeve. Therefore, it is important for athletes and coaches to maintain the position of the sensor as they throw to maintain accurate readings as movement of the sleeve from the intended sensor location will likely change the readings. In addition, the soft tissue artifact can lead to errors in readings for both the sensor and the aforementioned more popular sleeve, and previous literature has indicated the necessary use of data filtering techniques like the Kalman filter to eliminate much the error in readings (*Lin & Kulić, 2012*). It is possible that similar, more thorough filtering techniques would lead to even more precise Motus readings, as it is unclear what sort of filtering techniques the proprietary technology does use.

In addition, the smaller sample size still leaves questions as to the validity of the findings and the significant correlations did not always carry over across different pitch types: for example, the stress metric was significant in the off-speed pitch sample and not in the fastball pitch sample. Further research should be done with a larger sample size to both further investigate the arm speed metric in order to find a more intuitive significant correlation to a respective motion capture measurement and to further investigate the large numerical differences in the angular velocities of the two systems.

# CONCLUSION

This results from this study show that MotusBASEBALL could be a suitable low-cost and partial alternative to performing a full biomechanics capture, particularly for the arm slot, shoulder rotation, and stress metrics. Arm speed was shown to have a weaker correlation to the results that were found in the motion capture test. It should be noted that while all metrics from MotusBASEBALL had significant variance in values when compared to the motion capture metrics, the numbers were consistent for each subject and across all groups; arm slot averaged eight degrees (12.9%) less than motion capture, shoulder rotation averaged nine degrees (5.4%) less than motion capture, and stress averaged 41 (38.7%) and 42 Nm (39.3%) less than motion capture for elbow torque and shoulder torque, respectively. While differences in magnitudes prevented validation of the Motus scores, the high reliability of these three metrics in particular could reasonably be used in future studies and for use in monitoring an individual athlete's mechanics from session to session.

### Funding
The authors received no funding for this work.

### Competing Interests
All authors work at Driveline Baseball Enterprises, Inc.

### Author Contributions
- Kyle J. Boddy conceived and designed the experiments, analyzed the data, contributed reagents/materials/analysis tools, prepared figures and/or tables, authored or reviewed drafts of the paper, approved the final draft.
- Joseph A. Marsh performed the experiments, analyzed the data.

- Alex Caravan analyzed the data, prepared figures and/or tables, authored or reviewed drafts of the paper.
- Kyle E. Lindley performed the experiments.
- John O. Scheffey performed the experiments.
- Michael E. O'Connell performed the experiments, authored or reviewed drafts of the paper.

## Human Ethics

The following information was supplied relating to ethical approvals (i.e., approving body and any reference numbers):

Hummingbird IRB approved the study and granted ethical approval to carry out the data collection at the author's facilities (Hummingbird IRB #: 2018-10).

## Data Availability

Raw data are available in the Supplemental Files.

## Supplemental Information

Supplemental information for this article can be found online at http://dx.doi.org/10.7717/peerj.6365#supplemental-information.

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
