# Peer review of "Exploring wearable sensors as an alternative to marker-based motion capture in the pitching delivery"

_PeerJ, doi:10.7717/peerj.6365_

## Round 0.1 · original submission · Major Revisions

The three reviewers and I have read over your manuscript and are interested in the relative novelty of this data but have some quite major concerns at the moment. Due to the novelty of the data, we are happy to provide you the opportunity to revise this with respect to our comments. In particular, I would like you to particularly focus on the comments relating to aspects of your methodology; what metrics from the wearable sensors are actually exhibiting sufficient validity; and also the requirement to improve the overall quality of the scientific writing.

·

Basic reporting

Basic Reporting:

Thank you for inviting me to review this paper. I commend your taking this study on, as I believe the validation of these commercially available sensors is crucial. Much stock is placed in their produced data, and it's imperative that it is interpretable and valid in comparison to research-grade data. With an ever-expanding market of wearable devices, as well as the spectrum of their users, this is necessary. That an influential and notable training facility recognizes this and puts their own capital and labor into supplying the data is refreshing. I am glad to see this study come to review.
The authors present data supporting the reliability of wearable, Inertial Measurement Unit (IMU) sensors for the collection and analysis of biomechanical pitching data. They have spent the time needed to find suitable analogues between the two measurement systems (the wearable IMUs and the "gold standard" motion capture system) and are to be commended for finding a common framework in which to validate these sensors, which many practitioners use, but which aren't commonly validated in peer-reviewed literature.
Overall the paper is logical and well-written. The approach to validating the IMU sensors is well-constructed and I overwhelmingly agree with the way it's been undertaken. The need for this study is mostly well demonstrated in the prose. The collection of data is valid on its face. The statistical approach taken seems correct and (more importantly) is the proper one. Some of the terminology surrounding reliability and validity needs to be tightened, but overall this is a good paper. My review will be to offer suggestions for revision.

The approach I will take to this review will be to:

1. Comment on each facet of the review criteria in these sections (Basic Reporting, Experimental Design, Validity of the Findings, and General Comments for the Author(s)).
2. Make specific wording suggestions in the PDF itself via annotations.

Again, thank you for the invitation. I hope my comments are helpful and instrumental (pun not entirely unintended) to helping you produce the best possible study for publication. If I can clarify any comments, do not hesitate to contact me: travis.ficklin@dixie.edu.

Clear and unambiguous, professional English used throughout.

Yes, though there are multiple places throughout the manuscript where language could be tightened or clarified. I have made those comments in the annotated PDF directly. There are some spots where the word "valid" or a variant is used when reliability or consistency is meant, or is more accurate. Additionally, there are small grammatical things. None of the notations reflect fatal problems, so no more to mention here.

I will offer this suggestion, though: consider getting rid of any references to yourselves ("we", "us", "our", etc.) and replace them with generic, even passive language if you can. This is a personal preference on my part, and in the end, the editors style guide should rule.

Literature references, sufficient field background/context provided.

For the most part, I agree that it is. However, in demonstrating the need for this study (a "hole" in the existing literature), there are some studies missing. One in particular is very similar to yours, and so it needs to be included, with a rationale for why yours is still important. Here are some references to consider:

1. Augmented inertial measurements for analysis of javelin throwing mechanics - Sarkka, et. al., 2016, J. Sports Engineering. (This one seems to only make use of the sensor to monitor output kinematics and energetic of the javelin itself, so there is a big difference between it and yours - in fact, maybe this one isn't needed. I cannot insist you include it, but it is worth noting if you do.)
2. Validation of Inertial Measurement Units for Upper Body Kinematics - Morrow, et. al., 2017, J. App. Biomech. (Upper body kinematics, but applied to performing surgery - whoa - rather than to throwing. Again, huge difference between you and them, but this is a validation study, so it needs to be mentioned.)
3. A Distributed Wearable, Wireless Sensor System for Evaluating Professional Baseball Pitchers and Batters. - Lapinski, et. al., 2009, International Symp on Wearable Computers (IEEE). (This one is crucial to include - very similar to yours. Key differences: the motusBASEBALL system is already consumer-available and in widespread use - I am not sure their "SportSemble" is, and so there is a need for your validation right there. Also, their relationships are established only qualitatively, while their statistical analysis is a comparison (I think t-testing, but hard to tell from their text) to demonstrate that their sensor values are statistically equivalent to the MoCap values for *average* velocities.)

Professional article structure, figs, tables. Raw data shared.

Yes, and thank you for the raw data to peruse. The tables have some tightening that could be done, and I have annotated those in the PDF. That said, raw scatterplots of the values, with or without a line of slope=1 to visualize the correlations would be good to include. They tell a story in themselves.

Self-contained with relevant results to hypotheses.

Self-contained, yes. As to hypothesis:

1. Go ahead and include it in the introduction after your purpose statement (it's already in the abstract, so it's odd not to acknowledge it again).
2. You did hypothesize that the joint angle metrics would be accurate and reliable but not arm speed and stress metrics. You need to follow that up in your discussion. In the end, the IMUs are shown to be reliable, but not valid (consistent, but not accurate in comparison to "gold standard" values.)

Experimental design

Experimental Design:

Original primary research within Aims and Scope of the journal.

Yes. No further comment.

Research question well defined, relevant & meaningful. It is stated how research fills an identified knowledge gap.

This is addressed in my previous comment about the supporting literature. Your case is strong - just have to make it in light of the Lapinski study.

Rigorous investigation performed to a high technical & ethical standard.

Yes. I will add this: you're to be commended for getting the information from Motus to translate from their metrics to the MoCap metrics. They are to be commended for sharing information. Many companies will not, and they hide their algorithms as proprietary. Good on them and you for communicating and then disseminating.

Methods described with sufficient detail & information to replicate.

Mostly. I do think that the information is sufficient to replicate the study. The details of this can be tightened up some. Rather than address those points here, I will do so in the annotated text.

Validity of the findings

Validity of the Findings:

Impact and novelty not assessed. Negative/inconclusive results accepted. Meaningful replication encouraged where rationale & benefit to literature is clearly stated.

The strength of your study is the lack of an attempt to dismiss the information rendered by the Motus system (being that it isn't the same as what MoCap renders), but rather to put it into context of the information researchers are used to obtaining with their MoCap systems. You have shown nicely why the information available to consumers/practitioners is worthwhile, yet what grains of salt must be added. This is my awkward way of saying that your non-result result is useful when explained to practitioners. Please see my PDF comments for specifics on this line of thinking.

Data is robust, statistically sound, & controlled.

Yes. Except to say that data is a plural word, so data are robust, statistically sound, & controlled. I make this suggestion in you PDF too. If the Editor disagrees, go with his/her ruling!

Conclusion are well stated, linked to original research question & limited to supporting results.

I'll just state again that the paper will be strengthened by tying your conclusions/discussion directly back to your hypothesis. Additionally, please note any limitations your study has. For example, some of the reliability noted in your conclusion is limited to either all pitches en masse, or off-speed ones only - the stress metrics for fastballs were not strongly correlated enough to generalize the results to them. Also - note your sample pool. Are these collegiate athletes? High school? Pros? That has bearing on the generalizability of results. One last to mention: future work should be done on the day-to-day reliability of the typical use of the Motus sensor (sleeve). You state that it is consistent, but you either need to state that you have pilot data to support that, or that work should be done to establish it.

Speculation is welcome, but should be identified as such.

You really didn't make any speculations that seemed outlandish. The consistency of the sleeved use of the IMU needs to be addressed. See the paper for my note on that. In the end, the IMU is useful. The major caveat to that is that the values aren't really comparable to the MoCap ones. Rather, the usefulness is in monitoring changes in metrics (as training occurs, or as pitches are altered) - hence, reliable, but not valid. You do state this, but I would club the reader over the head with it.

Additional comments

General Comments to the Authors:

Again, good paper. I have made annotations to the PDF for you to use or not use. The purpose of my extended comments above is to document my rationale for voting for a "major revision". I think the paper should be published after the literature updates are added, and the tightening of the language is done. I realize that is right on the cusp between "minor" and "major", but I lean toward the latter because it will prompt one last systematic review before approval, and I feel that is important for any substantive revisions.

I wish you the best, and I hope to see a revised version soon to review again. I believe the changes will make it ready for publication.

Reviewer 2 ·

Basic reporting

Basic Reporting
1. Language and Writing: In general, the writing is very succinct and to the point. However, it therefore lacks depth and context for the study. The authors should consider the paragraph structure throughout. There are many very short paragraphs (and in some cases 1 sentence paragraphs in the discussion) that need refining.
2. The introduction and background are somewhat brief and lack depth in relation to wearables. The focus seems to be on just the baseball application. The authors would be well advised to provide more of a background on wearables (i.e. Chambers et al Sports Medicine July 2015, Volume 45, Issue 7, pp 1065–1081) and why they are necessary. This would ideally be placed before the baseball specific information. Generally, the paragraph structure in this section needs work. E.g. paragraphs 2 and 3 can be merged together.
Line 72 - There is literature out there that have looked at throwing and wearables that have not been included in this background presumably as its not baseball related. However, the study I believe is purely one of part validation of the system and thus other papers that have looked at throwing and wearables needs including. i.e. Wixted et al, Journal Sports Technology Volume 4, 2011 - Issue 3-4: Australian Sports Technology; Spratford et al, 2015 Journal of Sports Sciences Volume 33, 2015 - Issue 7;
I would also suggest in the intro and background that a section on the importance of product validation / comparison to gold standards is needed, as that in essence is what you are doing.
Line 84 needs a full stop at the end.

Figures:

Figure 1 – Geometric details of the sensor placement is needed – i.e. 5cm distal from the medial epicondyle etc etc – to ensure consistency in placement.
Figure 2 – is very dark (in black and white) and thus camera position can’t readily be seen.
Table 1 – Suggest centring the values in the table
Table 2- suggest centring the values in the column and in the row. As the speed measures are all compared to the same value – possibly merge the 3 rows on the Motus BASEBALL column with the same value to highlight that its just one value. Also, column spacing needs re-aligning.
Table 3 – this is unclear – more detail on what correlation is being assessed is needed. Difficult to interpret based on the table. Title of the table needs more detail.

Experimental design

Experimental Design
You state in the abstract a hypothesis yet in the background the contest of the main document this isn’t justified.
Furthermore, in the main document you do not state a hypothesis you only state a purpose. Please check Peer J requirements on this and ensure consistency.
Subjects: Can you provide some details on the years’ experience and the level of the pitchers? This would help contextualize the study further
As per the previous section I believe more context is required in justifying the research area.
Method Section Specifics
The details on the motion capture are well documented.
Line 116 - More detail on the Motus Baseball sensor is needed. Details relating to the sensor type etc will help the reader contextualize the results.
Important point: As the sensor was removed from the sleeve do you believe that would affect the results if it were in a sleeve? I understand why it has been removed.
Important Point: Your study design looks to compare the sleeve sensor to the capture system. Was though given to looking at the repeatability of the measures? Your subjects threw 5-7 fastballs – what was the within subject reliability of these measures? This would enhance the validity of the product. If you find that the sleeve underestimates (which looking ahead it did) then if it consistently underestimates compared to the gold standard measure, then we are able to comment on that when users wear the product. The key aim of a wearable should be to record data but then detect improvements or decrements in performance in its continued use. It is reliable this would assist in that narrative.
Line 144 – The Fleisig paper you reference doesn’t detail the methods used (its further references other papers). I would suggest including some detail on the methods here to allow replication by readers.
Line 181 – More detail is needed on what constitutes the stress measure from Motus.
Line 184-191 – suggest merging these paragraphs
Line 187-191 – You need to report what statistical package you used to calculate your statistics. I also think this study would benefit from some more robust statistics. Whilst correlations are valid, and should continue to be included, Bland Altman plots would really add some great value to this study. (more on the results in the next section).
Here you mention hypothesis again, but you haven’t stated a hypothesis outside of the abstract.
Line 200 – ‘while’ should be ‘whilst’
Line 202 – you have called them groups which implies they were separate subjects – you have used the same subjects for both throws thus I do not think groups is the right word. Suggest ‘condition’.
Table 4 – based on the power levels being below 0.8 for all but arm-slot how do you interpret these? I would also suggest this part of the analysis is associated with processing and statistical interpretation and thus should be included in the methods.

Validity of the findings

Validity of the Findings

1. Off speed arm extension metrics are greater that the fastball yet the correlation to ball speed, of which fastball was ~ 102 mph quicker were high? Can this be explained to help the reader understand this?
2. Table 3 – I found this difficult to interpret so I couldn’t comment in much detail. This needs refining and detail added.
3. You have found that the speed base metrics are grossly underestimated by the sleeve compared to the Opti track. Based on prior comments assessing if this underestimation is consistent may provide some useful information – i.e. if it consistently underestimates then a correction factor could be used to provide similar absolute values, but outside of that if it can detect changes that people could use it from performance improvements 9or decrements.)
Results specifics
Line 202 – you state that the FB group had ‘more’ significant p values – that is not how p-values work – its ether significant or not – it’s not necessarily a scale.

Discussion Specifics
Generally, this needs a lot of work. The authors have started by restating the main findings well (although again paragraph structure needs refining). However, there is minimal explanation of the results.
The discussion would benefit from assessing other papers that have looked at comparing measurement tools (i.e. O Donnell et al Measurement in Physical Education and Exercise Science
Volume 22, 2018 - Issue 1). There are many other sources out there comparing devices to accelerometers etc.
Line 255 – you state that the benefit of wearables is that they can be used in game – whilst this is undeniable, the validity of the method needs stating prior to that occurring. Thus, if the only measure that can be taken with some level of confidence is the arm slot – is it advisable to support its use by coaches? I would suggest further assessment is needed for this statement. The stress metric is unclear from the paper how it is calculated or what it is measuring
Conclusion – You mention variance within here and thus following on from previous comments it would be wise to assess this more formally.

Additional comments

The study is of interest as the world of wearables becomes increasingly popular in the sport and exercise fields to provide more information to relevant people on enhancing performance and potentially reducing injury. With that in mind one of the main flaws of wearables is there reliability and validity compared to gold standard methods, and thus if in fact the wearable is providing a useable measure. This study aims to do this and in part it has achieved this of which the authors should be congratulated. However there are a few flaws that I think need to be addressed that make the study for objective. Further statistical analysis and more interaction with the literature around validity and reliability of measurement systems is advised.

·

Basic reporting

The most important areas for the authors to consider in this manuscript in order of importance are the following:
1. Ensure the purpose statement is clear and sets the foundation for the methods that were chosen and the statistics that were performed. Currently the purpose is written different ways through the manuscript and the statistics only seem to partially address the intended purpose of the article.
2. The language of the manuscript should be re-visited extensively to ensure it is grammatically and scientifically less ambiguous.
3. Providing justification or potential speculation regarding the metrics that did not correlate should make up the majority of the discussion.
4. Visualisation of the correlation data should be performed to allow for analysis of the spread of the data as the r value can be deceptive.
5. Greater description related to the methods used for the optical motion capture (e.g. was it an existing validated model) and specific information regarding the elbow and shoulder, for example was it the same as the ISB recommendations: https://www.ncbi.nlm.nih.gov/pubmed/11934426) would ensure repeatability of the research.
6. Greater use of previous literature to support methods, justification of the discussion along with better use in the introduction (of prior validation studies of IMUs in comparison to “gold standard” could better set up the current manuscript and provide the authors examples of potential “best statistical practice” when seeking or performing a validation study.
Abstract
1. A notation of the language as an example: it is the data processing along with improvements in the hardware of IMUs that has led to their increased use and expanded measures that are purported to be possible with respect to kinematics and kinetics not simply “technological advancements in IMUs”. Re-visit language and choose to use descriptive biomechanical language (e.g. kinematics and kinetics) and specifics about the advancements (e.g. mixture of processing or programming with improved hardware) to ensure it is not presented as vague or ambiguous.
2. In hypothesis, “this device” – e.g. which device?
3. To state “…will be accurate and reliable” is better stated with acceptable reliability and accuracy as there is no definitive line of accurate and reliable as it is a continuous scale that is normally described qualitatively based on thresholds.
4. Ensure that the tense is appropriate, for example, in methods of abstract to state “will be” should be past tense as it is what they did. Further, methods statement in abstract could be combined to state more specifically. …breaking pitches with simultaneous assessment using motion capture and while wearing a MotusBASEBALL IMU. This would eliminate the need for the second sentence and be more accurate than stating “in the motion capture lab”
5. Results in abstract, first sentence is difficult to follow. Please re-visit. The term “test population” should be “sample”. The use of “very close” is not scientifically supported, ensure that is followed by the statistical output that supports it being “very close”.
6. Language of conclusion in abstract needs to be re-visited and ensure it is not a re-statement of that already stated in results.
Introduction:
1. Ensure the purpose statement at the end matches that used in the abstract and is specific enough to understand what the subsequent methods will require (particularly statistics). For example, to state “compare” typically infers that a group comparison will occur versus your purpose more surrounding validation and accuracy seems to be that what is intended.
2. Remove reference to the unpublished case study.
3. Remove the focus on your specific motion capture as the purpose is not about the type of motion capture it is the comparison of the single IMU (MotusBaseball; xxxx; xxx) which is how you should refer to instead of as the unit name, in comparison to the “gold standard”, optical motion capture (which is how you should refer to your system then just provide the (Unit Type, Manufacturer, Location).
Methods
1. Instead of “selected” state “volunteered
2. Merge this into a a single first paragraph, correct statement of what is in table 1 in line with below comment about “weight versus body mass”.
3. Was the model used previously validated or used in prior research?
4. With the IMU typically fixed with a compression garment, was any analysis performed to determine if this affected the measures (e.g. not having a compression over top?)
5. Offspeed = off-speed and state all the different types of off-speed pitches used.
6. What order butterworth filter?
7. As the position variables were taken at ball release, can it please be described if this is the same point at which the IMU measures are reported?
8. Name the app that is used to collect the IMU data specifically and move the equivalent descriptors of the variables provided by Motus up and would likely be advantageous to make a table that provides the descriptors of Motus versus optical motion capture descriptors for ease of comparison.
9. Need to re-visit the language used surrounding the purpose and subsequent statistics discussed.
10. State what was significance set at for statistical test, include confidence intervals. Consider addition of Bland Altman but this depends on better or more clear description of the purpose.
Results
1. Re-visit first lines 199-201 for clarity.
2. Avoid stating “more significant p-values” as a comparison of p values size unless this is intended to discuss number of significant variables but even then should be clarified.
3. Power analysis (post-hoc) is not as advantageous as providing the confidence intervals which should be added to the analysis (e.g. 95% CI for r values).
4. Alpha value should not be in results, should be in methods.
5. Avoid re-stating in-text the same things that are presented in the tables.
6. 219-223 – this information is not clear and can likely be totally removed.
Discussion
1. To improve the discussion as a whole it is critical that the authors re-visit and ensure that the expand not just re-state what was reported in the results. Further, one should re-visit the purpose/hypothesis if one states one as done in the abstract. Further, there should be more reference to prior research.
2. Greater justification and speculation would be useful with respect to the reason for such a difference between optical and IMU systems for the speed measures.
Tables
1. Table 1 could simply be entitled “Participant descriptive and performance characteristics” to reduce repeated listing of variables and state the subject number in here as (n = 10) remove from first line of table and appropriately name this Age (years). Further, these variables should be provided in metric but considering the sport it would then be advantageous to list the imperial commonly used in the sport. However, scientifically, the metric values should be reported. Further, the reported variables is body mass, not weight as the weight would be a force measured in Newtons. Further, to use the term off-speed here varies from in the abstract where the term used is “breaking” pitch.
2. Table 2 – ensure all abbreviations are also noted either in the title or below the table (preferable). For example, IR: Internal Rotation. Remember that figures and tables should be made in such a way they can stand alone from the manuscript. Further, clarify the varus stress location (e.g. elbow).
3. Table 3 – Please correctly label this as correlation coefficient (as an r) or explained variance which is r2. Further, in the table footnote, clarify what significance was set at (e.g. p < 0.05) and the variable names should be labelled identical to that in Table 2 to allow for easy comparison as it is currently listed, it is ambiguous as a standalone of what arm speed or what stress the results represent. Additionally, this table has plenty of room to provide confidence intervals (e.g. 95% for the r values) which will provide better information than post hoc power analysis listed in Table 4.

Experimental design

1. Ensure the study purpose and hypothesis statement clearly matches the methods and statistical analysis. For example, it first states “validate” but the hypothesis refers to accuracy and reliability and then the results return back to “valid” for casual use.
2. The experimental design is set up for a concurrent validation, but an extension of the statistical methods would allow for better support of statements. Including but not limited to: inclusion of confidence intervals, dependent samples t-test and/or bland-altman (which is a measure of agreement which would statistically match to the intention of “validation”). The reliability assessment used does allow for assessment of relative measure but to differentiate absolute versus relative reliability and further as stated to differentiate from agreement will allow you to better conclude the use (e.g. interchangeable or simply linearly correlated which means not interchangeable).

Validity of the findings

The results are valid but require more statistical analysis and greater reporting as suggested previously. The raw data seems appropriate. There is some justification for the authors to consider evaluating individual trials in addition to mean analysis as per suggestions from Mullineaux, D. R., Bartlett, R. M., & Bennett, S. (2001). Research design and statistics in biomechanics and motor control. Journal of Sports Sciences, 19(10), 739-760. doi:10.1080/026404101317015410 to determine if all trials should be used instead of the mean, especially considering the sampel only being n = 10.

Please see other comments regarding a clean link between purpose, methods, statistics, and conclusion.

Additional comments

1. The authors should be commended for seeking to perform validation on some of the commonly used IMU devices in the sport technology world. Such an effort should be applauded and is much needed. However, greater attention to statistical detail could further provide useful information for this validation attempt.
2. Please re-visit the manuscript for errors in capitalisation, use of parenthesis, method of statistical reporting and overall grammar (e.g. tense, ambiguity, passive voice) is re-visited.
3. Ensure that full development of paragraphs occur within the introduction, though the flow is logical, more depth surrounding the measures to be evaluated should be considered (e.g. the measures of arm slot, arm speeds and varus elbow stress and shoulder IR stress should be introduced).
4. Label the data more consistently prior to publication for open source

---

## Round 0.2 · Minor Revisions

Thanks for addressing most of the initial concerns with the manuscript. There are just a few small ways in which it needs further corrections before it can be accepted for publication.

·

Basic reporting

This revision of the paper is much improved, and the vast majority of my previous comments and/or concerns have been address satisfactorily. I recommend some minor revisions before acceptance. The updates to the manuscript to clarify statistical procedures and place the findings in the context of other published research make the study stronger. Many of the comments I include here are carried from my first review of the paper. The minor revisions I recommend are noted in comments on the PDF I am uploading. If they do not appear or are unclear, please let me know.

The authors present data supporting the reliability of wearable, Inertial Measurement Unit (IMU) sensors for the collection and analysis of biomechanical pitching data. There is better explanation of the sensors and the metrics included by the manufacture in their software in this revision. Additionally, better discussion is made as to differences between the systems and the limitations imposed by using IMUs.

The paper remains overall logical and is now better-written. There are changes, additions, and clarifications made in the manuscript made in response to reviewers. Once or twice, the changes include explanatory notes which are not necessary to the final reader. I have noted those in the PDF as well. All that said, I commend the authors for their thoughtful responses to review, and look forward to the publication of the paper after minor edits.

Please find in my review:

1. Comments on the review criteria (Basic Reporting, Experimental Design, Validity of the Findings, and General Comments for the Author(s)). These comments center on your changes made in response to my first review. They are brief.
2. A PDF file with recommended edits as annotations/comments. These have much more detail as to wording, etc. If these comments are not appearing, please alert me – I will find another way to make them.

Clear and unambiguous, professional English used throughout.

The writing is much tighter this time – nice work. I made some comments in the PDF for your consideration – or that of the editor. The paper reads more professionally this time. First person references are removed, and passive, scientific voice is now prevalent.
Literature references, sufficient field background/context provided.

The authors have now provided a more complete background for the study. The included references and their salient points in now present and leads logically to the need for their study. I have no further concerns about this part.
Professional article structure, figs, tables. Raw data shared.

The figures and charts are much better. The data plots add much better visualization. Thank you for adding them.

Self-contained with relevant results to hypotheses.

The hypothesis now appears in conjunction with the purpose statement in the introduction, which is much better. The discussion about the reliability of the Motus IMU vs. its validity (or where its validity is lacking compared to “gold standard” motion capture) is much better. I find it to be clear and reflective of the data that are presented.

Experimental design

Experimental Design:

Original primary research within Aims and Scope of the journal.

Yes. No further comment.

Research question well defined, relevant & meaningful. It is stated how research fills an identified knowledge gap.

This is now satisfied. Particularly, the explanation of important differences between the Lapinski study and yours are crucial to bolstering the case for publishing your findings.

Rigorous investigation performed to a high technical & ethical standard.

Yes – no concerns.

Methods described with sufficient detail & information to replicate.

The authors have used reviewer comments to clarify and strengthen their methods. Filtering techniques, instrumentation techniques and placements, and statistical procedures are all clearer. I believe researcher could follow and replicate the study’s procedures.

Validity of the findings

Validity of the Findings:

Impact and novelty not assessed. Negative/inconclusive results accepted. Meaningful replication encouraged where rationale & benefit to literature is clearly stated.

No concerns.

Data are robust, statistically sound, & controlled.

No major concerns.

Conclusion are well stated, linked to original research question & limited to supporting results.

Added discussion of limitations is good. Explanation of generalizability of results is improved, and the tying of results to research question and hypothesis is clear. No further concerns.

Speculation is welcome, but should be identified as such.

N/A

Additional comments

General Comments to the Authors:

This is a vastly improved paper. You are to be commended for your attention to detail in responding to my comments and those of the other reviewers. I have made some last comments/annotations on the uploaded PDF that have to do with voice and/or grammar. My major concerns are addressed, and I recommend the paper be published once those minor changes are made. I commend the authors for their professional approach to this review process and look forward to further manuscripts from their work.

---

## Round 0.3 · Minor Revisions

General comments

I feel you have addressed the comments of the reviewers adequately, but I have a few minor revisions requested before the manuscript will be accepted for publication.

Specific comments

Abstract: please report the actual correlation values in the results section of the abstract.
Introduction, paragraph 5, line 3: please write this as “from Camp et al., (2017).
Methods, paragraph 4: please provide more detail about the MotusBASEBALL sensor here or elsewhere in regards to some of the specifics of the sensor. For example, provide some detail on its size and mass, sample rate, sensitivity as well as the actual types of sensors (accelerometers, magnetometers and gyroscopes) contained within the unit.
Overall Discussion: I normally find it useful to have both absolute and relative errors reported in studies of this nature. Therefore, when you report a difference between the two systems in the angular displacement, angular velocity or other metrics, please report these as an absolute difference (using the standard unit of measurement) as well as a percentage difference to assist the reader better understand the magnitude of these differences.
Limitations, paragraph 1: you mention the use of a different sleeve as a limitation. Can you please be more specific here and earlier in the manuscript about the reason for this and how this may result in some oscillation of the IMU with respect to the underlying skin, muscle and bone? Further, can you discuss how this potential movement artefact in such a high-speed activity could be reduced by the use of a different sleeve, different components within the sensor or data filtering approaches; and how this may contribute to improved absolute validity of the unit?

·

Basic reporting

No concerns at this point.

Experimental design

No concerns at this point.

Validity of the findings

No concerns at this point.

Additional comments

The article is cleaned up and ready to publish, in my opinion. Congratulations on a good study, and I look forward to more.

-TKF

---

## Round 0.4 · accepted · Accept

Thanks for addressing our concerns